



# Underestimation of global O₂ loss in optimally interpolated historical ocean observations

Takamitsu Ito[1], Hernan E. Garcia[2], Zhankun Wang[2], Shoshiro Minobe[3,4], Matthew C. Long[5], Just Cebrian[2,6], James Reagan[2], Tim Boyer[2], Christopher Paver[2], Courtney Bouchard[2], Yohei Takano[7], Seth Bushinsky[8], Ahron Cervania[1], Curtis A. Deutsch[9]

[1]School of Earth and Atmospheric Sciences, Georgia Institute of Technology, Atlanta, Georgia USA
[2]NOAA, National Centers for Environmental Information, Silver Springs, Maryland, USA
[3]Department of Natural History Sciences, Graduate School of Science, Hokkaido University, Sapporo, Japan
[4]Department of Earth and Planetary Sciences, Faculty of Science, Hokkaido University, Sapporo, Japan
[5]Climate and Global Dynamics, National Center for Atmospheric Research, Boulder, Colorado, USA
[6]Northern Gulf Institute, Mississippi State University, Stennis Space Center, Mississippi, USA
[7]Los Alamos National Laboratory, Los Alamos, New Mexico, USA
[8]School of Ocean and Earth Science and Technology, University of Hawaii at Manoa, Honolulu, Hawaii, USA
[9]Department of Geosciences, Princeton University, Princeton, NJ, USA

*Correspondence to*: Takamitsu Ito (taka.ito@eas.gatech.edu)

**Abstract.** The global ocean's oxygen content has declined significantly over the past several decades and is expected to continue decreasing under global warming with far reaching impacts on marine ecosystems and biogeochemical cycling. Determining the oxygen trend, its spatial pattern and uncertainties from observations is fundamental to our understanding of the changing ocean environment. This study uses a suite of CMIP6 Earth System Models to evaluate the biases and uncertainties in oxygen distribution and trends due to sampling sparseness. Model outputs are sub-sampled according to the spatial and temporal distribution of the historical shipboard measurements, and an optimal interpolation method is applied to fill data gaps. Sub-sampled results are compared to full model output, revealing the biases in global and basin-wise oxygen content trends. The optimal interpolation underestimates the modeled global deoxygenation trends, capturing approximately two-thirds of the full model trends. North Atlantic and Subpolar North Pacific are relatively well sampled, and the optimal interpolation is capable of reconstructing more than 80% of the oxygen trend. In contrast, pronounced biases are found in the equatorial oceans and the Southern Ocean, where the sampling density is relatively low. Optimal interpolation of the historical dataset estimated the global oxygen loss of 1.5% over the past 50 years. However, the ratio of global oxygen trend between the subsampled and full model output, increases the estimated loss rate to 1.7 to 3.1% over the past 50 years, which partially overlaps with previous studies. The approach taken in this study can provide a framework for the intercomparison of different statistical gap-fill methods to estimate oxygen content trends and its uncertainties due to sampling sparseness.



# 1 Introduction

Historical observations indicate that the ocean oxygen ($O_2$) inventory has declined in recent decades, a trend that has been termed ocean deoxygenation (Keeling et al., 2010; Levin, 2018). Ocean heat uptake causes the reduction of oxygen solubility, and changes in ocean circulation and biogeochemical processes. Ocean warming and increasing stratification can further decrease $O_2$ exchange between upper and deep layers, further reducing the oceanic $O_2$ inventory. The reduction of dissolved oxygen can have far-reaching impacts on marine habitats (Deutsch et al., 2015; Gruber, 2011; Pörtner & Farrell, 2008; Vaquer-
Sunyer & Duarte, 2008).

The distribution of historical $O_2$ measurements is irregular and sparse. The calculation of changes in the global $O_2$ content requires filling the data gaps in time and space, making it difficult to quantify global trends and their uncertainties. Recent estimates of the global oxygen decline are in the range of 0.5-3.3% (IPCC, 2022) relative to climatological means over the
period of 1970-2010 (Helm et al., 2011; Ito et al., 2017; Schmidtko et al., 2017). The wide range in the estimates of ocean deoxygenation can stem from different interpolation methods to estimate global $O_2$ content, different data quality control standards, and different data sources. Previous studies estimating the rates of ocean deoxygenation have relied on World Ocean Database 2018 (WOD18) (Boyer et al., 2018). WOD18 is an international collaboration among national data centers, oceanographic research institutions and investigators to provide a comprehensive dataset of quality-controlled oceanographic
variables. Shipboard observations are more prevalent in the northern hemisphere oceans in the warm seasons. Oxygen measurements from a single year (e.g. 1991; Fig. 1A) do not adequately cover the global ocean; a pentadal composite (e.g. 1989-1993; Fig. 1B) is performed to increase the coverage at the expense of averaging out the high-frequency variability on the timescale shorter than 5 years. Even so, there are large data gaps in the South Pacific and Indian Ocean. In such a case, optimal interpolation (hereafter, OI) has been widely applied to fill data gaps and yield a gridded data field (Fig. 1C), which
produces the best-fit $O_2$ distribution in the least square sense given the covariance structure in the dataset (Wunsch, 1996). One shortcoming of OI application is that it can underestimate $O_2$ trend in data-sparse regions. Without any measurements nearby, the mapped field approaches asymptotically to the climatology (i.e. to zero oxygen anomaly). If there is a widespread $O_2$ decline but only a fraction of ocean volume is sampled, the OI will underestimate the declining trend of ocean $O_2$ content.




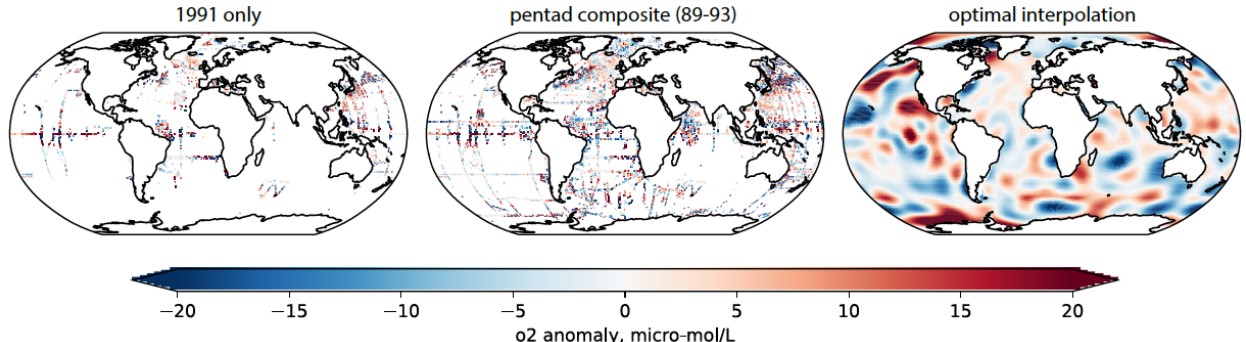

**Figure 1.** Maps of (left) single year observation for the $O_2$ anomaly in Year 1991 at 200m depth, (middle) the pentadal composite $O_2$ anomaly centered at Year 1991 covering from 1989 to 1993 at 200m depth, and (right) the optimally interpolated pentadal $O_2$ anomaly based on the data in the middle figure.


The objective of this study is to use a suite of Earth System Model (ESM) simulations as a testbed to evaluate the uncertainties in ocean deoxygenation rates by sub-sampling model output according to the spatial and temporal distribution of the historical shipboard measurements. Earth system models represent our current understanding of physical and biogeochemical processes expressed in mathematical equations. These processes and their interactions are numerically integrated forward in time, predicting the trajectory of the Earth's climate system. ESMs generate their own natural variability that reflects chaotic behavior of the natural climate system, but its temporal trajectory does not necessarily match that of the real world. Observed $O_2$ changes may be influenced by both external forcing (such as volcanism and anthropogenic greenhouse gases and aerosol emissions) and natural climate variability. These models are imperfect and often include varying degrees of biases due to inadequate process understanding and the lack of computational resources to resolve critical processes at smaller length/time scales. Current earth system models do not fully reproduce the $O_2$ variability and trends (Oschlies et al., 2017; 2018), and observational data is essential to validate the model output. In turn, the analysis of model output can inform the range of underlying variability and trends.

This study uses seven different ESMs from Coupled Model Intercomparison project phase 6 (CMIP6) that provided dissolved oxygen output. These seven models sample the range of $O_2$ variability and trends that can arise from different model architectures, biogeochemical parameterizations and modes and phases of natural climate variability. Globally gridded $O_2$ fields from ESMs provide fully-sampled states, and thus perfectly known model trends, for the simulated variables. The modelled $O_2$ distribution can also be sub-sampled according to the time-evolving pattern of historical ocean observations, to evaluate the effect of sampling sparseness. We purposefully remove information from the model output where there was no *in situ* measurement. This hypothetical "observation" of model output, with its realistic data gaps, can be used to evaluate the uncertainties in ocean deoxygenation rates due to both data sparseness and statistical gap-filling approaches. The sub-sampled



model output can then be subject to statistical gap-filling method (OI) to evaluate how well the fully-sampled states can be reconstructed. It is of great interest to evaluate to what extent the OI method underestimates the true $O_2$ trend in the context of the simulated deoxygenation.


The structure of this paper is as follows. The second section describes the analysis method, data sources and the earth system models. The third section describes the results, followed by the interpretation of the results and conclusion in section four.

## 2 Methods

### 2.1 Observational data source

We make use of observations from the bottle and Conductivity-Temperature-Depth instruments (CTD) $O_2$ data in WOD18. Dissolved oxygen is the third most frequently measured chemical tracer in the ocean, following temperature and salinity. There are approximately 2.8 million temperature, 2.4 million salinity, and 0.9 million $O_2$ vertical profiles in the Ocean Station Data (OSD, or simply bottle data) reported in WOD18. In addition, CTD data includes approximately 1 million temperature and salinity profiles, and 0.2 million $O_2$ profiles. The OSD (i.e. bottle) $O_2$ data are largely located on the margins of the ocean

basins and along repeat hydrographic transects (Figure 2). $O_2$ observations in the OSD profile were typically measured by modified "Winkler titration" method with a precision of about 1 µmol/kg (Carpenter, 1965). Most modern oxygen chemical titration measurements are based on Carpenter's whole bottle titration method and an amperometric or photometric end-detection with a precision of about 0.5-1µmol/kg (or approximately, 0.3%). The CTD-$O_2$ data are based on electrochemical and optical sensors mounted on the CTD-rosette samplers, which are periodically calibrated to the Winkler $O_2$ (Gregoire et al.,

2021). The coverage of CTD measurements increased after the 1990s and that of profiling floats rapidly increased in recent years. However, the overall spatio-temporal coverage of $O_2$ observations from bottle and CTD has decreased since the 1990s. Profiling floats $O_2$ data increased significantly in the past 10 years, however, its precision is on the order of 1-2% (~2µmol/kg) and its data quality control and calibration is still under development especially in the upper ocean oxycline (Bittig et al., 2018; Maurer et al., 2021), thus float $O_2$ data have been excluded in this study.




**Figure 2.** Number of OSD and CTD oxygen profiles aggregated into 1°x1° longitude-latitude grid cells for four decades from 1965 to 2014. The color scale indicates the number of measurements in log scale.

## 2.2 Data pre-processing and optimal interpolation

The pre-processing of the data includes a check for data quality where only acceptable data using the WOD18 quality control (QC) flags. The original WOD18 standard-depth profiles with 102 depth levels are placed into bins which are the 1°x1° longitude-latitude grid cells with 102 vertical depth levels referenced to the standard depths of WOD18. Of the 102 vertical depth levels, 47 levels are in the upper 1,000m.






The target analysis period is after 1965 when the modern oxygen titration method was established by Carpenter as referenced above. Some of the data from most recent years are not included in the ESMs as discussed below, so the analysis ends in 2014. The spatially binned quality-controlled data were averaged at monthly resolution where mean, variance and sample size are recorded from 1965 to 2014 for the bottle data, and from 1987 to 2014 for the CTD-O$_2$ data. Next, the monthly mean

climatology is determined by calculating the climatological monthly mean combining the bottle and CTD-O$_2$ data and then filling data gaps. We are interested in long-term O$_2$ changes which can be calculated as the anomalies from the monthly climatological mean. Departures from the monthly climatology are recorded as O$_2$ anomalies for each bin. The binned data is very sparse at monthly timescale (Figure 1). For each year, the monthly anomaly data is averaged into yearly anomalies neglecting the months with missing data. This step increases spatial data coverage significantly while averaging out high-

frequency variability in the data including changes shorter than the yearly timescale such as waves and eddies. In addition, a 5-year moving window (pentadal) averaging is applied to the yearly anomaly neglecting the years with missing data. This further increases the spatial data coverage, while averaging out variability on the timescale shorter than 5 years. The resulting, pentadal O$_2$ anomaly data covers the 46-year period from 1967 to 2012.

The optimal interpolation (OI) is applied to the pentadal O$_2$ anomaly data for each year to yield the spatially interpolated O$_2$ anomalies following Wunsch (1996). This method provides the least-square estimate of O$_2$ field on regularly spaced grid cells, minimizing the mean square error of the mapped data for given observations with a covariance function. Stationary and isotropic Gaussian covariance is assumed throughout this study, with the e-folding length scale ($L_{ref}$) of 1,000 km. This particular choice of length scale controls how far an observation can influence the far field together with the assumed noise-

to-signal ratio ($\varepsilon$) of 0.2. The Gaussian assumption may be qualitatively reasonable, but the ocean circulation is neither spatially stationary nor uniform. The use of Gaussian function allows us to avoid calculating and storing the large and complex covariance structure but it can distort the resulting maps (Fukumori et al., 1991), which is a caveat for this study. A basin mask is used to interpolate data points only within the same ocean basin such as the Atlantic, Pacific, Indian and Southern Ocean. Each 1°x1° grid point is assigned to one of the 53 basins defined in the Appendix 1 of *Garcia et al., (2019)*. The binned

oxygen vector, **X**, is expressed as a (N x 1) vector where N is the number of binned data for a particular basin. The objective map of oxygen climatology, **Y**, is a (M x 1) vector, where M is the number of grid cells for the basin. The optimal interpolation is applied to each basin as follows.

$$\mathbf{Y = DE^{-1}X} \tag{1}$$


Where **X** is the pendadal oxygen anomaly input from the discrete data, and **Y** is the objective map of (gap-filled) oxygen anomaly. **D** is a (M x N) data-grid covariance matrix based on the Gaussian function, where $D_{mn} = exp(-L_{mn}^2/L_{ref}^2)$ and $L_{mn}$ is the distance between the two points. **C** follows the same definition but for the N x N data-data covariance, and $\mathbf{E = C} + \varepsilon\mathbf{I}$ where $\varepsilon$ is the noise to signal covariance ratio. For the Southern Ocean, all data points southward of 30°S are used. An example





of this process in Year 1991 is shown as Figure 1. Basin-wise application of optimal interpolation is performed for the $O_2$ anomalies resulting in yearly (running pentadal) maps for the 46-year period. The $O_2$ anomaly field as well as its standard error field are recorded.

## 2.3 Ocean deoxygenation trend

Using the yearly maps of the $O_2$ anomaly field, global and basin-wise $O_2$ content are calculated as the volume integral over
the upper 1,000m, $O(t)$, where $t$ is time since 1967. The magnitude is referenced to the mean value of the first 10 years where the 10-year (1967-1976) mean $O_2$ contents are subtracted from respective $O_2$ content time series for comparison purposes. Ocean deoxygenation trends are estimated as the slope ($a$) of the $O_2$ content time series using standard linear regression.

$$a = \frac{\sigma_{tO}}{\sigma_{tt}} \qquad (2)$$

$$b = \overline{O} - a\overline{t} \qquad (3)$$

where $\sigma_{tO}$ is the covariance between time and $O_2$, and $\sigma_{tt}$ is the variance in time. ($a,b$) are slope and intercept of linear regression. Assuming that the regression errors are normally distributed, the standard error for the slope ($\epsilon_a$) and intercept ($\epsilon_b$) can be calculated as follows.

$$\epsilon_a = \sqrt{MSE\left(\frac{1}{\sum(t_n - \overline{t})^2}\right)} \qquad (4)$$

$$\epsilon_b = \sqrt{MSE\left(\frac{1}{N_{eff}} + \frac{\overline{t}^2}{\sum(t_n - \overline{t})^2}\right)} \qquad (5)$$

where *MSE* stands for the mean square error of regression, $t_n$ is time at n-th data point. The gridded $O_2$ dataset is constructed based on 5-year running mean. An effective sample size ($N_{eff}$) is calculated assuming that 5-year data are independent, thus $N_{eff} \sim 9$ for 46 years of data. These parameters are later used to evaluate the uncertainty and will be used for the comparison between models and observation.

## 2.4 CMIP6 Earth System Models

Two sets of time-varying $O_2$ fields are derived from the ESMs including the full field and the reconstructed model output (Table 1). We selected a subset of earth system models participating in the Coupled Model Intercomparison Project Phase 6 (CMIP6), and their outputs for the historical simulation are downloaded from the Earth System Grid Federation (https://esgf.llnl.gov). The monthly mean $O_2$ output is first re-gridded onto the global 1°x1° longitude-latitude grid for the period of 1965 to 2014. The bilinear interpolation is first performed for the horizontal interpolation, followed by the linear
interpolation on the vertical axis to the standard depths of the WOD18. Sub-sampled model output is then generated from the full field where the model output fields were resampled using the same spatial and temporal locations as the observations. Similar to the observational analysis, the "sub-sampled" monthly $O_2$ climatology is assembled from the sub-sampled data with the optimal interpolation filling the data gaps using Eq (1). Then $O_2$ anomalies are calculated by subtracting the "sub-sampled"



monthly climatology, and they are first aggregated into annual $O_2$ anomalies neglecting months without data, followed by the
running pentadal averaging. Finally, the basin-wise optimal interpolation is applied to yield the reconstructed $O_2$ anomaly
fields using Eq (2). These procedures are repeated for each of the models in Table 1. For the comparison purposes, the 5-year
moving window averaging is applied to the full field.

| Model name | Variant | Reference |
|---|---|---|
| CanESM5 | r1i1p1f1 | Swart et al (2019) |
| MPI-ESM1-2-LR | r1i1p1f1 | Mauritsen et al (2019) |
| GFDL-ESM4 | r1i1p1f1 | Dunne et al (2020) |
| IPSL-CM6A-LR | r1i1p1f1 | Boucher et al (2020) |
| MIROC-ES2L | r1i1p1f2 | Hajima et al (2020) |
| NorESM2-LM | r1i1p1f1 | Seland et al (2020) |
| E3SM1-1 | r1i1p1f1 | Burrows et al (2020) |

**Table 1.** List of CMIP6 models used in this study. Variants represent decadal-scale variability ensemble members, and are
coded according to (r) realization, (i) initialization, (p) physics, and (f) forcing. The first available variant, typically noted as
r1i1p1f1, is taken from each model.

## 3 Results

### 3.1 Observed and modelled $O_2$ trend maps

The observational trend is first determined based on the optimally interpolated gap-filled WOD18 profiles. The vertically
integrated $O_2$ inventory (0-1000m) trend pattern is shown in Figure 3. While regional differences exist, the basin-scale patterns
of the observed $O_2$ loss are similar to previous studies [*Helm et al.,* 2011*; Ito et al.,* 2017*; Schmidtko et al.,* 2017]. In the North
Atlantic, overall $O_2$ decline is observed except for the south of Greenland in the subpolar North Atlantic where a patch of an
increasing trend exists. In the North Pacific, a strong decrease is found the western subpolar region spreading from the sea of
Okhotsk (Nakanowatari et al., 2007), which may be connected to the reduced ventilation in this region. A weak increase is
found in the subtropical North Pacific (Ito et al., 2019), which is related to the multi-decadal natural variability of the North
Pacific climate. Oxygen increases are observed in the subtropical southern hemisphere oceans and in the south of Greenland.
In terms of the global inventory trends, the data suggests a global linear trend of -175 +/- 24 $TmolO_2$/decade, or approximately,
1.5% loss over the 50-year period.

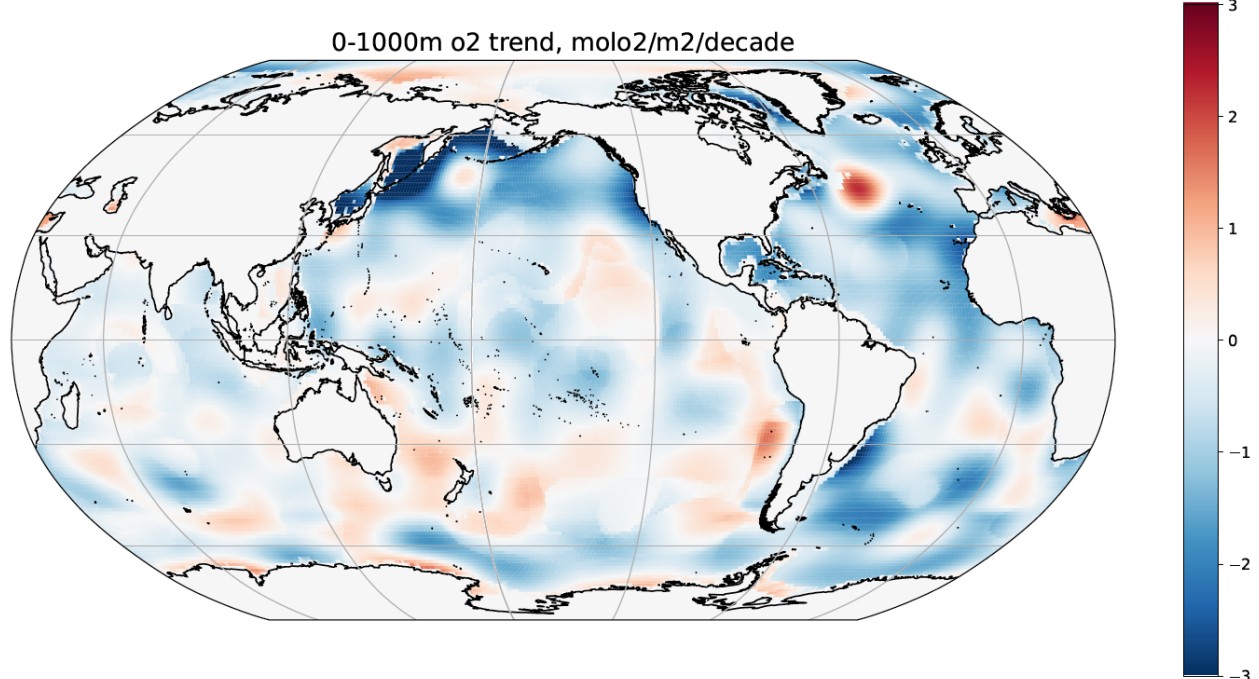

**Figure 3.** Linear trend of upper ocean (0-1,000m) column $O_2$ inventory from 1967 to 2012 from the optimally interpolated pentadal $O_2$ anomaly based on the World Ocean Database 2018.

210

Supplementary Figures (S1 and S2) show the comparison of the trend pattern between the observation and models listed in Table 1 for the full model field (Figure S1) and the reconstructed model output (Figure S2). The modeled $O_2$ trend patterns are moderately correlated to the observations for some of the CMIP6 models as summarized in Table 2. CanESM5, MPI-ESM1-2-LR, IPSL-CM6A and MIROC-ES2L exhibit a moderately positive correlation of approximately r=0.3. It is interesting to

215    contrast this result to the hindcast simulation of the earlier generation of ocean biogeochemistry model reported by *Stramma et al.* [2012]. The subset of CMIP6 models analyzed in this study are slightly better correlated to observational estimates than the hindcast runs using earlier generation of models. This is likely due to the improved biogeochemical model structure and parameterization rather than the physical climate forcing. Hindcast simulations are forced by the observed atmospheric variability through the meteorological reanalysis products. In contrast, historical simulations of the CMIP6 models generate

220    natural climate variability that does not always capture the phasing of observed variability.





| | Subsampled model R with WOD2018 | Full model R with WOD2018 | Full and subsampled R | Global trend, Tmol/dec (sub-sampled/OI) | Global trend, Tmol/dec (full) |
|---|---|---|---|---|---|
| CanESM5 | 0.34 | 0.32 | 0.63 | -89 | -165 |
| MPI-ESM1-2-LR | 0.3 | 0.27 | 0.58 | -77 | -115 |
| GFDL-ESM4 | 0.03 | 0.04 | 0.37 | -60 | -120 |
| IPSL-CM6A-LR | 0.31 | 0.25 | 0.69 | -74 | -82 |
| MIROC-ES2L | 0.24 | 0.22 | 0.45 | -40 | -64 |
| NorESM2-LM | 0.09 | 0.13 | 0.53 | 4 | -45 |
| E3SM1-1 | 0.06 | 0.08 | 0.53 | -23 | -44 |

**Table 2.** From left to right column, spatial pattern correlation (Pearson's correlation coefficients) between observed and modeled upper ocean (0-1,000m) column $O_2$ trend and the global trend magnitudes, the pattern correlation between observed modeled $O_2$ trend patterns full and subsampled optimally interpolated model outputs.

The reconstructed CMIP6 model output is slightly better correlated to the observation than the full model output for the majority (5 out of 7) of models, perhaps reflecting the common sampling pattern and gap-filling approach. Comparing the reconstructed and the full field from the same model, the pattern correlation of the $O_2$ trend ranges from 0.37 to 0.69. While it is not perfect, the OI can estimate the general pattern of the full field with moderate correlation for the 1°x1° gridded trend maps. This motivates us to further investigate to what extent OI can estimate the $O_2$ trend for a larger scale, hemispheric and global domain.

## 3.2 Global and hemispheric $O_2$ inventory time series

The globally integrated $O_2$ content has a stronger declining trend than in ESMs, and the weak trend bias in models becomes even greater when reconstructed from subsampled data (Fig. 4; Table 2). Only one of the models exceeded the observed global trend in full field (CanESM5, -165Tmol/decade; Table 2). When there are no observations nearby, the OI reverts to the background climatology, thus decreasing the amplitude of anomalies. Thus, the estimated $O_2$ content tends to underestimate $O_2$ anomalies in the region of sparse sampling. The magnitude of underestimation depends on the distance from observations which sets the covariance according to the assumed Gaussian function. Figure 4 further shows that the sub-sampling introduces three decadal-scale peaks in Years 1988, 2000 and 2011 for both observations and some of the models (Figure 4b). These quasi-decadal peaks are not apparent in the full model output (Figure 4a). We hypothesize that these quasi-decadal peaks are likely spurious, caused by the sparse sampling pattern.

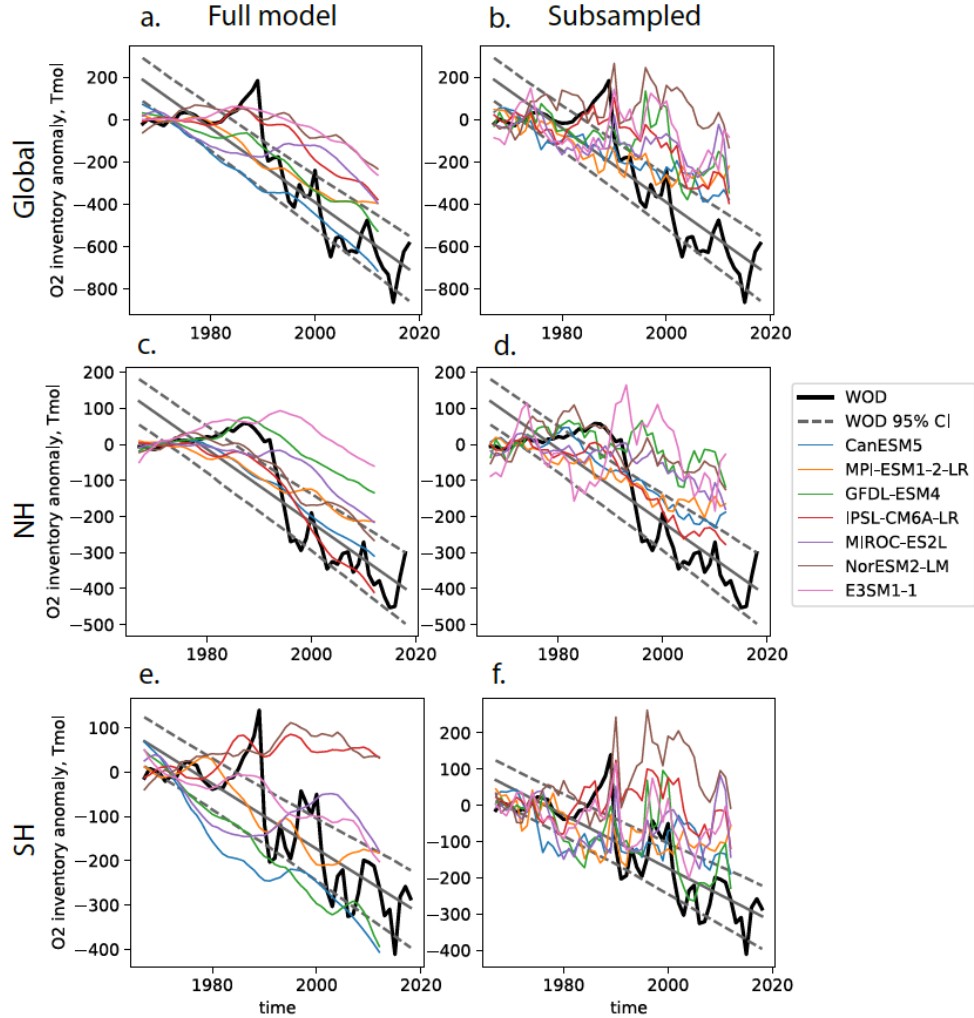

**Figure 4.** Time series of upper ocean (0-1,000m) column $O_2$ inventory from World Ocean Database 2018 (black solid) and CMIP6 models (color) from (left panels) full model output and (right panels) subsampled and optimally interpolated model output. Top row is the global, and the middle and bottom rows are the northern and southern hemispheres. The vertical axis for the hemispheric inventories is half of the global inventory. Gray lines are the linear regression with 95% confidence interval calculated from the Monte Carlo method with randomized slope and intercept using the uncertainties from equations (3-6). The inventory anomaly is referenced to the first ten-year averages.

The global inventory time series is divided into the northern and southern hemispheric components (Figure 4; middle and bottom rows). Comparing the hemispheric and global inventory time series indicates some notable issues with the sub-sampling. First, northern hemispheric trends in some of the full model output (IPSL-CM6A-LR and CanESM5) have similar





magnitudes to the observational trends. In some other ESMs, the overall magnitudes of the southern hemispheric trends are similar to the observation (GFDL-ESM4 and CanESM5). Overall, the hemispheric trends are in similar magnitudes between the north and the south for observations and models. The reconstructed model outputs appear to underestimate the magnitude of the trends for all models. The magnitude and the causes of this underestimation are of great interest and will be investigated further in the following sections.

Secondly, the observed quasi-decadal peaks primarily appear in the southern hemispheric inventory (Figure 4ef, solid black line), and some of the models reproduce these peaks (GFDL-ESM4, MIROC-ES2L, NorESM-LM2, E3SM1-1) for the reconstructed model output (Figure 4f). There are no apparent peaks in the full model output, confirming that these features are spurious.

Thirdly, in the northern hemisphere (Figure 4cd), there is a moderate increase towards the late 1980s and then it decreases strongly during the 1990s. Two of the earth system models (GFDL-ESM4 and E3SM1-1) show similar increase in the early period (Figure 4c), however, they underestimate the decreasing trend after the 1990s. These features are distorted in the reconstructed model output. It is difficult to determine whether the apparent increase of oxygen content is meaningful during the 1980s, but similar features are found in earlier studies focusing on the near surface waters [*Garcia et al.,* 2005].

The models and observations tend to disagree more significantly in the southern hemisphere. Modeled inventory trends disagree substantially from one another because of the spurious quasi-decadal noises. While some models exceed the observed magnitude of oxygen decline (GFDL-ESM4, CanESM5), some other models even show increases in the southern hemispheric $O_2$ inventory (NorESM2-LM, IPSL-CM6A-LR).

## 3.3 Basin-wise $O_2$ inventory trends

To examine $O_2$ trends across ocean basins, we divided the global data into 13 regions according to a basin mask (shown in Figures 5-6). The basin $O_2$ inventories are integrated for each region, from the observations (upper left) and the full model output (Figure 5) and sub-sampled model (Figure 6). Blue color shading shows strong $O_2$ loss, and red indicates $O_2$ increase. The regional $O_2$ trends from the reconstructed model output are displayed in Figure 6. The effect of subsampling does not change the spatial pattern but it only affects the trend magnitudes. As expected, the reconstructed model output exhibit weaker trend magnitudes. For each region, the inventory time series are displayed in separate figures from supplementary Figure S4 through S16. For the basin-scale deoxygenation trend, the North Atlantic Ocean is the only basin where all models show the same sign of change relative to the observation for the full field and reconstructed model output (Figure 5 and 6).





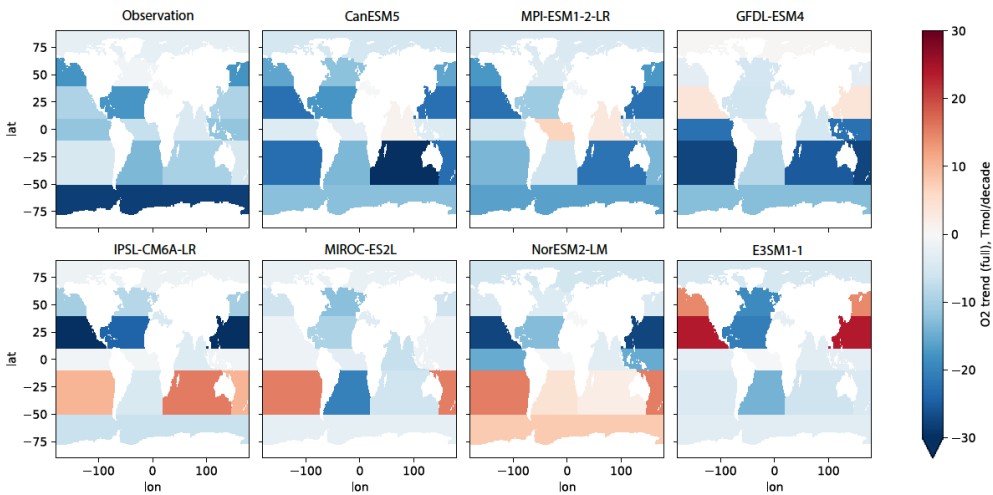

**Figure 5.** Observed and modeled basin-wise linear trend of the $O_2$ inventory from 1967 to 2014 from the observations and full model output. The global ocean is divided into 13 regions including Subpolar North Atlantic (SPNA), Subtropical North Atlantic (STNA), Equatorial Atlantic (EQAT), Subtropical South Atlantic (STSA), Mediterranean Sea (MED), Subpolar North Pacific (SPNP), Subtropical North Pacific (STNP), Equatorial Pacific (EQPA), Subtropical South Pacific (STSP), Equatorial Indian Ocean (EQID), Subtropical South Indian Ocean (STSI), Southern Ocean (SO), and Arctic Ocean (AO).

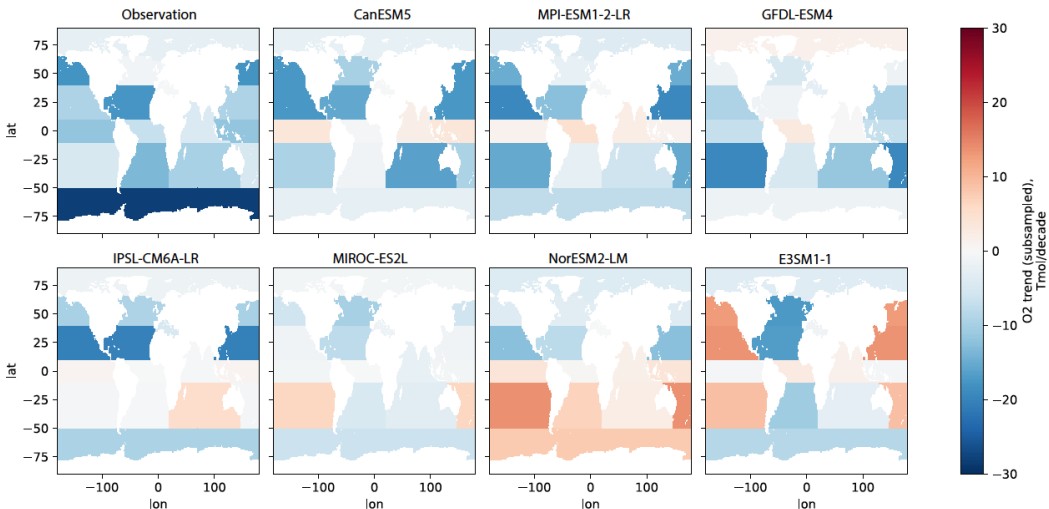

**Figure 6.** Same as Figure 5 but for the trend reconstructed from optimal interpolation of sub-sampled model output.


Figure 7 shows the evolution of the spatial data coverage for each basin. To calculate the percent coverage value, the number of grid cells with at least one shipboard profile is divided by the total number of grid cells in each basin. Overall, the North
Atlantic and Mediterranean Ocean are the most well observed among the 13 regions. Near surface waters are better sampled than the deeper layers (400m, 700m). The data coverage evolves over time, depending on the basin. During the 1970s and 80s, there was greater data coverage for near surface waters (100-200m), and the near-surface data coverage gradually decreased after the 1980s. However, this pattern is not uniform through the depths. For some regions such as the Subpolar North Atlantic, there appears to be no significant decrease in deeper profile (700m).


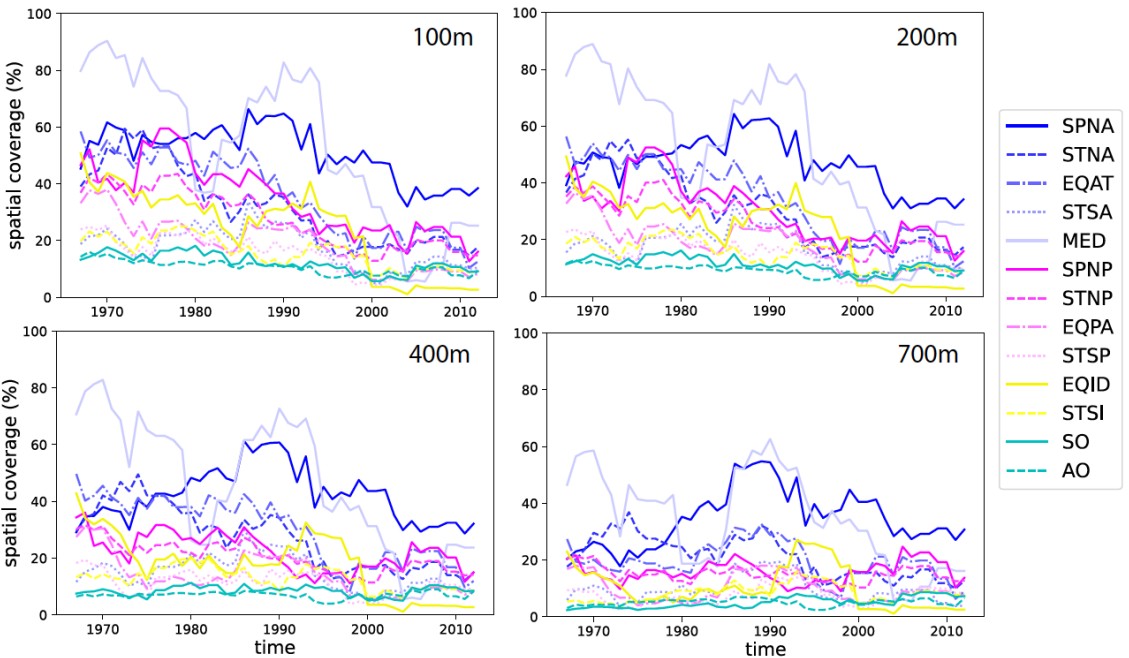

**Figure 7.** Spatial data coverage of the pentadal $O_2$ anomaly data from WOD18. The number of grid cells with at least one profile is divided by the total number of cells for each basin. Blue lines are Atlantic basins. Magenta lines indicate Pacific basins. Indian basins are in yellow, and cyan is used for the Southern Ocean and the Arctic Ocean. Sub-basins are coded by
the intensity of each color. Abbreviations for the basin names are as follows. Subpolar North Atlantic (SPNA), Subtropical North Atlantic (STNA), Equatorial Atlantic (EQAT), Subtropical South Atlantic (STSA), Mediterranean Sea (MED), Subpolar North Pacific (SPNP), Subtropical North Pacific (STNP), Equatorial Pacific (EQPA), Subtropical South Pacific (STSP), Equatorial Indian Ocean (EQID), Subtropical South Indian Ocean (STSI), Southern Ocean (SO), and Arctic Ocean (AO).




There are several notable features from this comparison. First, models exhibit varying patterns of $O_2$ changes, and the model-disagreements are more pronounced in the southern hemisphere oceans even in the full model output. This is consistent with the varying hemispheric-scale trend magnitude as shown in Figure 4ef. Approximately half of the models show increasing/decreasing trends in the Subtropical South Pacific. The observed $O_2$ decline is strong in the observations, but its

time series is noisy and the discrepancies between the full field and the reconstructed model output are large in the Southern Ocean (Figure S15). This is consistent with the persistently low data coverage in the Southern Ocean (Figure 7). The Southern Ocean contributes significantly to the spurious quasi-decadal peaks that are visible in the hemispheric and global time series (Figure 4), thus the observed trend in the Southern Ocean may include large uncertainty.

There are two regions, the Subpolar and Subtropical North Atlantic, that observations and all models agree in the sign of changes. These two region's inventory time series are displayed as supplementary Figure S4 and S5. In the Subpolar North Atlantic, the magnitude of modeled $O_2$ changes bracket the observation where some models (CanESM5, IPSL-CM6A-LR, E3SM1-1) exhibit even stronger $O_2$ loss than observations. In the Subtropical North Atlantic, these three models exhibit a similar magnitude of $O_2$ loss as the observations. In the equatorial Atlantic, there is a clear difference between the models and

observation. The observation shows a decreasing trend and not a single model was able to reproduce it. Similarly, the models were not able to reproduce the magnitude of $O_2$ loss in the subpolar North Pacific with the exception of MPI-ESM-1-2-LR.

**3.4 Synthesis**

The basin-wise $O_2$ trend is compared between the full field and reconstructed model output in Figure 8, assessing the ability of the OI method to reproduce the full-field data. In Figure 8, the horizontal axis is the full model and the vertical axis is the

reconstructed model output. Each dot indicates simulated $O_2$ trend magnitude for a basin. The red solid line is the 1:1 ratio, indicating that the OI method was able to fully reproduce the trend magnitude. Most of the dots are located between the red solid line and the purple dash line, indicating that the magnitude of the ocean deoxygenation trend is underestimated due to the OI method applied to sparsely sampled data.






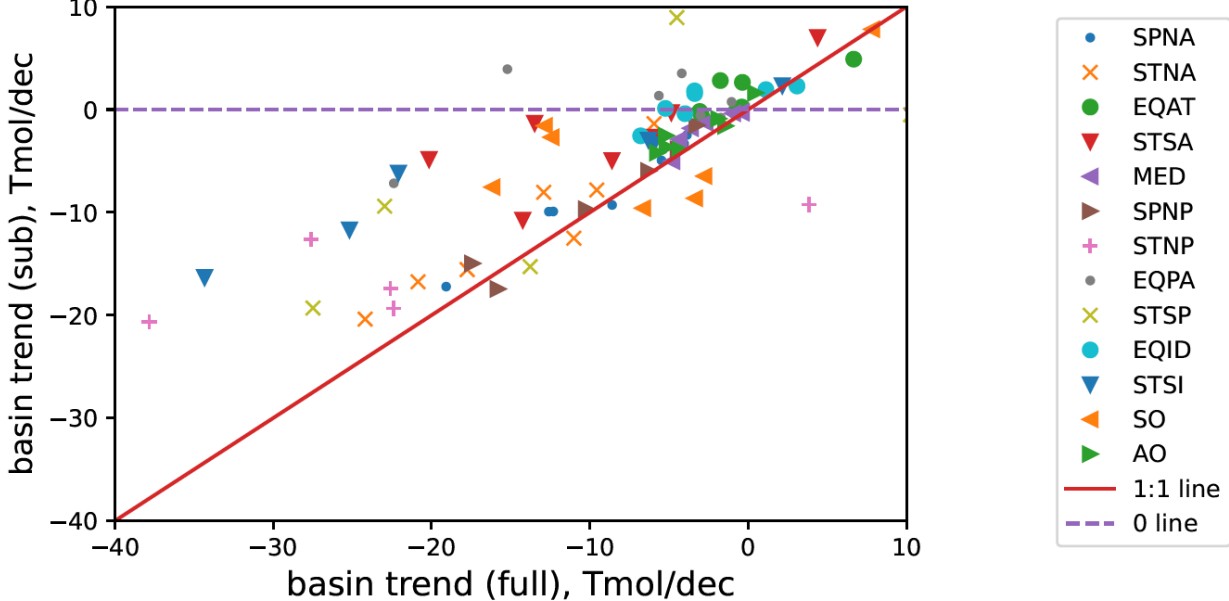

**Figure 8.** Basin-wise relationship between fully sampled and sub-sampled $O_2$ trend for seven CMIP6 models. Data points on or near 1:1 line (red solid) indicate that sub-sampled data adequately reproduced the fully sampled modeled trend. Abbreviations for the basin names are as follows. Subpolar North Atlantic (SPNA), Subtropical North Atlantic (STNA), Equatorial Atlantic (EQAT), Subtropical South Atlantic (STSA), Mediterranean Sea (MED), Subpolar North Pacific (SPNP), 350 Subtropical North Pacific (STNP), Equatorial Pacific (EQPA), Subtropical South Pacific (STSP), Equatorial Indian Ocean (EQID), Subtropical South Indian Ocean (STSI), Southern Ocean (SO), and Arctic Ocean (AO).

Four regions (Subtropical North Atlantic, Subpolar North Atlantic, Mediterranean, Subpolar North Pacific) performed very well in terms of capturing more than 80% of the deoxygenation trend in the context of the simulation. These regions are 355 relatively well sampled and the loss of the trend magnitude due to the OI is minimal. In contrast, the four regions (Equatorial Atlantic, Equatorial Pacific, Equatorial Indian, Southern Ocean) performed very poorly capturing less than 30% of the simulated deoxygenation trend. These regions unfortunately are not well represented by the subsampled and gap-filled data, showing the limitation of the OI method. The strong negative trend in the Southern Ocean (upper left panel in Figure 5) may be highly uncertain, and this is concerning since the Southern Ocean significantly contributes to the global oxygen content. 360 Other basins (Subtropical South Atlantic, Subtropical North Pacific, Subtropical South Pacific, Subtropical Indian Ocean, Arctic Ocean) are moderately represented (30-80% of the true trend).

*To what extent did the OI method underestimate the global deoxygenation trend?* Figure 9 illustrates the relationship between the "true" global trend (as x-axis) and the "estimated" trend from the sub-sampled model output. Each dot comes from a model 365 from two different ways of aggregating the global trend. The blue dots include all basins regardless of the ability of the OI




method to reconstruct the "true" trend. The purple dots exclude the equatorial basins as well as the Southern Ocean. The linear regression among the 7 models informs that the sub-sampling and the gap-filling with the OI method can capture approximately two-thirds (68%, purple line) of the "true" trend excluding the low confidence regions. Looking at the distribution among the models, the spread of this ratio is 19% as calculated by the standard deviation. If all basins are included, the fraction that is

retrieved by the OI method decreases to 58% (blue line).

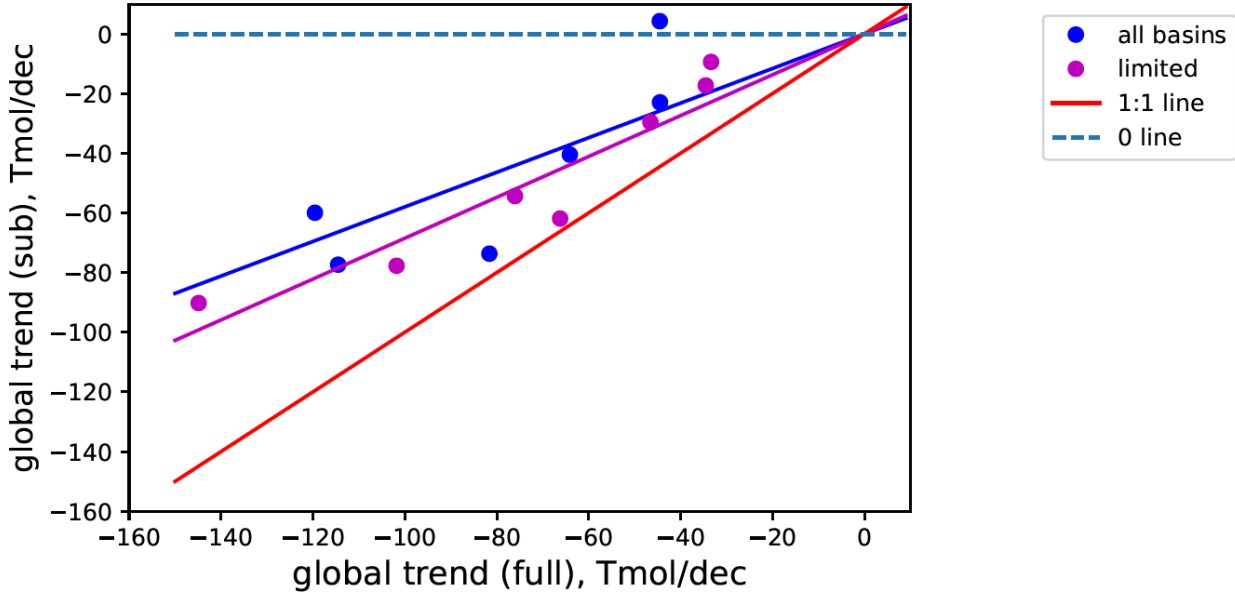

**Figure 9.** Global relationship between fully sampled and sub-sampled model $O_2$ trend. Blue dots indicate the 7 CMIP6 models with full global model output. Purple dots indicate the same except that the four poorly represented regions (Equatorial

Atlantic, Equatorial Pacific, Equatorial Indian, Southern Ocean) are excluded.

One of the implications from the model analysis is that the optimal interpolation method used in this study may result in the significant underestimation of the dissolved oxygen trend in observations. The observation-based global oxygen content trend can be adjusted assuming that the ratio of deoxygenation trend between the sub-sampled and full model output is approximately

two-thirds (68±19%) as determined by the CMIP6 ESMs. Optimal interpolation of the WOD18 oxygen profiles estimated a 1.5% $O_2$ decline over the last 50 years, but the true $O_2$ decline may be in the range of 1.7 to 3.1%. This partially overlaps with the recent estimates of the global oxygen decline which is in the range of 0.5-3.3% (IPCC, 2022), but suggests that the low end of that range is very unlikely.



## 4 Discussion

The premise of this study is that earth system models can provide useful information about the uncertainties in global ocean deoxygenation rate due to the sparse sampling and the specific gap-filling method used. The models disagree amongst each other and with the observations because of different and imperfect representation of processes due to model structures, parameterizations, and the presence of natural variability. However, a model can estimate the observational sampling bias by comparing its "true" model state to one that is reconstructed by sub-sampling model output according to the pattern of

shipboard profiles (bottle and CTD) from the WOD18.

The model-based analyses in this study are generally consistent in showing that subsampling with the gap-filling method yields weaker trends than the full model output. The gap-fill method used in this study is optimal interpolation (OI), which provides the "best-fit" distribution of $O_2$ anomaly in the least square sense assuming the Gaussian covariance structure of the data. The

OI method essentially predicts a diminishing anomaly when there is no observation nearby with the assumed e-folding length scale. If there is a wide-spread $O_2$ decrease, the OI can underestimate the trend in a sparsely sampled region. Our result confirmed this tendency for the global deoxygenation trends from the subset of CMIP6 earth system models. Our analysis, based on 7 such models, suggests that approximately two-thirds of the "true" trend is captured by the reconstructed model output. This conclusion generally applies to all models independent of the model skills to capture the observed trend for the

global and hemispheric inventories (compare left and right column of Figure 4) and for the basin-wise trends (compare Figure 5 and 6). Some ocean regions have better coverage than others, and significant regional variations exist for the sampling density and thus the performance of OI. For example, the North Atlantic and subpolar North Pacific are relatively well sampled, and the OI was able to capture more than 80% of the "true" trend. In these well-sampled regions, detailed analyses of ocean deoxygenation rates are likely fruitful using models and observations using the OI method.


Broadly speaking, the northern hemisphere oceans are generally better sampled than the southern hemisphere oceans, but the overall trends appear to be equally contributed by both hemispheres (Figure 4). Basin-wise analysis revealed diverging basin-wise trend patterns among the models (Figure 5 and 6). There is no consistent pattern in the contributions from different regions to the overall trend regardless of the sampling density. Also, there is no consistent pattern in the sign of multi-decadal $O_2$ trends

except for the North Atlantic Ocean, where all models are in general agreement. This region has the highest sampling density (Figure 7) and the full-field and reconstructed $O_2$ trends are in good agreement (Figure 8). Data coverage is not the only factor, but it plays an important role for the performance of the OI method. The North Atlantic is sampled at 20-50% density based on 1° x 1° grid cells with decreasing coverage from the surface to deeper depths and from subpolar to subtropical latitudes (Figure 7 and S17). In the low-sample region, namely, the Southern Ocean whose data coverage is persistently less than 13%,

the OI method struggled to reconstruct the full-field $O_2$ trends. In this region, historical observations are limited to certain





longitudes/latitudes (e.g. Drake Passage) and the repeat hydrographic cruises (Figure 2), and it was clearly inadequate to represent the full-field data.

It is useful to compare oxygen content trends using multiple gap-filling approaches to assess the uncertainties from different methodology (IPCC, 2022). The framework developed in this study may be helpful to further deepen such intercomparison studies and to quantify the skill of different gap-filling methods in the context of model output. Such comparison study may reveal what sampling density is sufficient to reconstruct the real trend. For such exercise, it is important to select model-derived oxygen fields that include realistic background variability. For the OI method, it is also crucial to have the covariance structure of $O_2$ field. An important caveat for this study is that the models used here were not eddy-resolving, and we also used a Gaussian covariance with a prescribed length scale. Mesoscale ocean eddies are energetic features at the spatial scales of 10-100km and timescales of several months. The model outputs did not include this type of internal ocean variability, and the modeled fields did not include the mesoscale "noises" that are present in the observation. The use of non-eddying models reduces the level of internal ocean variability much lower than the observations. Thus, we are not able to address to what extent the trend estimates vary depending on the presence of ocean eddies and smaller scale variability, which is a caveat in this study. It may be possible to emulate the mesoscale eddy "noise" and to estimate a more realistic covariance structure of $O_2$ fields using outputs from detrended high-resolution simulations, but this is beyond the scope of this paper and is left for future study.

While sampling sparseness is likely a major source of uncertainty, there are other sources of uncertainties that remain open for further investigation. Historical $O_2$ profiles may have evolving precision and uncertainty that are difficult to replicate in a model-based study. For example, a Winkler titration performed on a Nansen bottle during the 1960s may have different precision than a more recent Winkler titration done on a Niskin bottle using amperometric or photometric end-point detection methods. Looking ahead, integration of autonomous float $O_2$ data will pose challenges in terms of assessing uncertainties that are changing with the evolution of measurement techniques.

**Acknowledgement**

The oceanographic data that comprise the WOD have been acquired through many sources and projects as well as from individual scientists. In addition, many international organizations such as the IODE/GODAR and WDS have facilitated data exchanges, which have provided much data to the WOD. TI is grateful for funding support from Department of Energy (DE-SC0021300), National Science Foundation (OCE-2123546), and JSPS International Fellowship for Research in Japan. SM is supported by the Japan Society forthe Promotion of Science (JSPS) KAKENHI (JP18H04129). YT is supported as part of the Energy Exascale Earth System Model project, funded by the U.S. Department of Energy, Office of Science, Office of Biological and Environmental Research. The submission has been approved and assigned LA-UR number LA-UR-23-20603.





**Code/Data Availability**

The World Ocean Database (WOD) is available from NOAA National Center for Environmental Information. CMIP6 model outputs are available from Earth System Grid Federation.

**Author contributions**

All authors developed the ideas of this study, and TI designed the methods of data analysis and performed most of the analysis. The manuscript was mainly written by TI with major comments and revisions by CD, SM, MCL, HEG, ZW, and YT. All authors contributed to the article and approved the submitted version.

**Competing interests**

The authors declare that the research was conducted in the absence of any commercial or financial relationships that could be construed as a potential conflict of interest.

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
