# Peer review of "Underestimation of multi-decadal global O2 loss due to an optimal interpolation method"

_Biogeosciences, 2023_

## Author Comment (AC1)

September 9th, 2023

Author Comments to Reviewer #1

We appreciate the encouragement and thoughtful comments. Below we provide point-to-point responses in red font.

The authors have presented an impressive analysis of interest to how global ocean oxygenation levels are changing. I believe that this study will be of broad interest for not only the detection of climate change, but also in motivating further community research activities. My recommendation is that the study be accepted for publication after only relatively small/minor changes that are detailed below.

Line 220:

Where the authors say "...that does not always capture the phasing of observed variability", can they say instead: "nature climate variability that in general does not reproduce the phasing of observed variability"?

We appreciate the suggestion. The sentence in L220 will be revised accordingly and it will flow much better.

Fig. 4b:

Can the authors point out how much larger these spurious signals (mapping error) are than Pinatubo etc.?

This is an excellent suggestion for comparison, giving an intuitive sense of mapping uncertainty. A recent modeling study (Fay et al., 2023) demonstrates that the effect of Pinatubo caused cooler SST and increased uptake of oxygen following the eruption in 1991. While the study was based on a single ESM large ensemble (CESM1-LENS), it showed over 80TmolO2 increase in O2 inventory within a few years, which is comparable to the amplitudes of the variability for both observation and models shown in Fig 4b. The manuscript will be revised around L245 on this point.

Given that the authors have collective familiarity and experience in working with large ensembles, even without doing any additional analysis, how would natural variability uncertainty measure up against any of the stories emphasized here?

We think that the contribution of natural variability is very likely important, but it is also difficult to quantify with the existing observations or a collection of single runs from the CMIP archive unfortunately. As suggested, the best approach would probably be to use multi-model large ensembles with adequate ensemble members of randomized natural climate variability.

The recent study mentioned above (Fay et al., 2023) indeed showed some evidence of the important contributions from natural variability seen as a spread in their Figure 2c. While it's based on a single model, it supports potentially crucial role of natural variability. We think this helps our concluding discussion and the source of uncertainties other than the sampling sparseness around Line 435.

And again without needing to perform additional analyses, it would be good if the authors can comment in a few sentences on the relative importance of anomalies in AOU versus O2SAT in determining the observed trend in O2 for the real ocean. As a related question, if globally extrapolation/mapping were to be performed using AOU and O2 separately on density horizons, would that make a difference?  Or even if O2 itself were to be mapped on density surfaces, do the authors believe that this other aspect of mapping is an issue in producing spurious errors?

This paper has focused on mapping and the trends of total $O_2$ and it is good to refer to the two components, O2sat and AOU. Surface waters are relatively close to the saturation with overlying atmosphere. In general AOU is close to zero in the surface water, thus O2sat plays a dominant role at the surface.  The importance of AOU increases with greater depth.  If global mapping were to be performed separately for O2sat(S,T) and AOU, the mapping of O2sat would essentially reflect that of temperature with some minor contributions from salinity.  Since temperature has been measured with much higher sampling rates, its mapping uncertainty would be significantly lower than that of O2/AOU.

On the second point, distributions of temperature, density and O2sat are known to co-vary. It would make sense to horizontally interpolate O2 (or AOU) and O2sat on density horizons for at least two reasons. First, temperature variation is much smaller on density surfaces than on z-levels, so the O2sat variation would be better constrained on isopycnals. Secondly, ocean transport in the interior ocean is primarily oriented along isopycnal surfaces, thus, it is very likely that the interpolation (smoothing) on isopycnal surfaces would reduce spurious errors. On the other hand, there could be technical difficulties. Winkler O2 measurements come from sparse bottle samples, and the sampling depths unlikely resolve the locations of the desired isopycnals. The calculation of inventory trends would also depend on the accuracy of isopycnal thickness.  In the end, one would have to try and evaluate whether or not and how

much uncertainty can be reduced by mapping isopycnally. While it is beyond the scope of this study, we think it would be a promising topic for future study to compare O2sat/AOU inventories calculated in depth and isopycnal coordinates and more importantly their impacts on uncertainty. The section 4 would be a good place to add this discussion around L435.

Minor editing points:

Throughout the text, the authors should replace "northern hemisphere" by "Northern Hemisphere", I believe, to comply with the convention (same for the Southern Hemisphere)

We appreciate the suggestion. If we are allowed to revise the manuscript, the text will be corrected accordingly.

Also, I believe that "Earth system model" should be used instead of "Earth System Model".

We appreciate the suggestion. If we are allowed to revise the manuscript, the text will be corrected accordingly.

Line 48: change to:

"WOD represents an international collaborative effort among…"

We appreciate the suggestion. If we are allowed to revise the manuscript, the text will be corrected accordingly.

Lines 56-57: change from "Without any measurements nearby…" to

"For regions without any nearby measurements…"

We appreciate the suggestion. If we are allowed to revise the manuscript, the text will be corrected accordingly.

Line 58: change to

"… the OI method will underestimate the declining trend…"

We appreciate the suggestion. If we are allowed to revise the manuscript, the text will be corrected accordingly.

Line 76:  change "to valid the model" to

"For the evaluation of the model…"

We appreciate the suggestion. If we are allowed to revise the manuscript, the text will be corrected accordingly.

Line 175: change "reconstructed model output" to

"…reconstructions fr subsampled model output"

We appreciate the suggestion. If we are allowed to revise the manuscript, the text will be corrected accordingly.

Line 177: change "…outputs for the historical simulations to…" to

"…outputs for their historical simulations to…"

We appreciate the suggestion. If we are allowed to revise the manuscript, the text will be corrected accordingly.

Line 179: change "The bilinear" to "A bilinear"

We appreciate the suggestion. If we are allowed to revise the manuscript, the text will be corrected accordingly.

Line 181: Change "…as the observations" to "..as with the observations"

We appreciate the suggestion. If we are allowed to revise the manuscript, the text will be corrected accordingly.

Line 198: Change "...similar to previous studies..." to "..similar to those in previous studies..."

We appreciate the suggestion. If we are allowed to revise the manuscript, the text will be corrected accordingly.

Line 203: change "...in the south of Greenland" to "...to the south of Greenland"

We appreciate the suggestion. If we are allowed to revise the manuscript, the text will be corrected accordingly.

Line 330: change "There are two regions, the Subpolar and Subtropical..." to "There are two regions, namely the Subpolar and Subropical..."

We appreciate the suggestion. If we are allowed to revise the manuscript, the text will be corrected accordingly.

Line 332: change "bracket the observation where some models" to "bracket the observations whereas some models..."

We appreciate the suggestion. If we are allowed to revise the manuscript, the text will be corrected accordingly.

Line 341:  change "indicating that the OI method" to "indicating where the OI method"

We appreciate the suggestion. If we are allowed to revise the manuscript, the text will be corrected accordingly.

Line 386:  replace "gap-filling method used" with "gap-filling method used with observations"

We appreciate the suggestion. If we are allowed to revise the manuscript, the text will be corrected accordingly.

Line 394: replace "least square sense assuming the Gaussian" with "least squares sense assuming a Gaussian..."

We appreciate the suggestion. If we are allowed to revise the manuscript, the text will be corrected accordingly.

Line 396: replace "wide-spread" with "widespread"

We appreciate the suggestion. If we are allowed to revise the manuscript, the text will be corrected accordingly.

Lines 425-426:  replace "at the spacial scales of 10-100km and timescales of several months" with

"With characteristic spatial scales of 10-100km and characteristic timescales..."

We appreciate the suggestion. If we are allowed to revise the manuscript, the text will be corrected accordingly.

This is the end of our responses. Thank you again for the thoughtful comments.

---

## Author Response (AR1)

Georgia Tech College of Sciences
School of Earth and Atmospheric Sciences

**School of Earth & Atmospheric Sciences**
Atlanta, Georgia 30332-0340 U.S.A.
PHONE: +1.404.894.0231
FAX: +1.404.894.5638

November 16th, 2023

*Dear editorial board of Biogeoscience,*

*We appreciate this opportunity to submit the revised manuscript. We followed the review comments to perform a major revision. The majority of the figures are significantly modified from the original submission (Figs. 3, 4, 5, 6, 7) and the text has been edited throughout. Below, we present point-by-point responses to the review comments and we explain our corresponding revisions. Below, the original reviewer's comments are presented in black and our responses are in blue fonts. Please note that the line numbers appear significantly different between the "track change" and plain version of the manuscript. The line numbers used in this response letter are matching with the "track change" version of the revised manuscript.*

**Reviewer 1**

The authors have presented an impressive analysis of interest to how global ocean oxygenation levels are changing. I believe that this study will be of broad interest for not only the detection of climate change, but also in motivating further community research activities. My recommendation is that the study be accepted for publication after only relatively small/minor changes that are detailed below.

Line 220:

Where the authors say "…that does not always capture the phasing of observed variability", can they say instead: "nature climate variability that in general does not reproduce the phasing of observed variability"?

 The sentence is revised accordingly and it flows much better (now, in Line 260 of the revised manuscript).

Fig. 4b:

Can the authors point out how much larger these spurious signals (mapping error) are than Pinatubo etc.?

This is an excellent suggestion for comparison, and giving an intuitive sense of mapping uncertainty. A recent modeling study (Fay et al., 2023) demonstrates that the effect of Pinatubo caused cooler SST and increased uptake of oxygen following the eruption in 1991. While the study was based on a single ESM large ensemble (CESM1-LENS), it showed over 80TmolO2 increase in O2 inventory within a few years, which is comparable to the amplitudes of the variability for both observation and models shown in Fig 4b. A sentence is added to L287 on this point.

Given that the authors have collective familiarity and experience in working with large ensembles, even without doing any additional analysis, how would natural variability uncertainty measure up against any of the stories emphasized here?

We think that the contribution of natural variability is very likely important but is also difficult to quantify with the existing observations or a collection of single runs from the CMIP archive unfortunately. As suggested, the best approach would probably be to use multi-model large ensembles with adequate ensemble members of randomized natural climate variability. The recent study mentioned above (Fay et al., 2023) indeed showed some evidence of the important contributions from natural variability seen as a spread in their Figure 2c. While it's based on a single model, it supports potentially crucial role of natural variability. We think this helps our concluding discussion and the source of uncertainties other than the sampling sparseness around Line 623.

And again without needing to perform additional analyses, it would be good if the authors can comment in a few sentences on the relative importance of anomalies in AOU versus O2SAT in determining the observed trend in O2 for the real ocean. As a related question, if globally extrapolation/mapping were to be performed using AOU and O2 separately on density horizons, would that make a difference? Or even if O2 itself were to be mapped on density surfaces, do the authors believe that this other aspect of mapping is an issue in producing spurious errors?

This paper has focused on mapping and the trends of total $O_2$ and it is good to refer to the two components, O2sat and AOU. If global mapping were to be performed separately for O2sat(S,T) and AOU, the mapping of O2sat would essentially reflect that of temperature with some minor contributions from salinity. Since temperature has been measured with much higher sampling rates, its mapping uncertainty would be significantly lower than that of O2/AOU.

On the second point, distributions of temperature, density and O2sat are known to co-vary. It would make sense to horizontally interpolate O2 (or AOU) and O2sat on density horizons for at least two reasons. First, temperature variation on an isopycnal is much smaller than that of O2, so the O2sat variation would be better constrained on

isopycnals. Secondly, ocean transport in the interior ocean is primarily oriented along isopycnal surfaces, thus, it is very likely that the interpolation (smoothing) on isopycnal surfaces would reduce spurious errors. On the other hand, there could be technical difficulties. Winkler O2 measurements come from sparse bottle samples, and the sampling depths unlikely match the exact location of the desired isopycnals. The calculation of inventory trends would also depend on the accuracy of isopycnal thickness.  In the end, one would have to try and evaluate whether or not and how much uncertainty can be reduced by mapping isopycnally. While it is beyond the scope of this study, we think it would be a promising topic for future study to compare O2sat/AOU inventories calculated in depth and isopycnal coordinates and more importantly their impacts on uncertainty. The section 4 would be a good place to add this discussion, and the sentences are added around L603.

Minor editing points:

Throughout the text, the authors should replace "northern hemisphere" by "Northern Hemisphere", I believe, to comply with the convention (same for the Southern Hemisphere)

The text has been corrected accordingly.

Also, I believe that "Earth system model" should be used instead of "Earth System Model".

The text has been corrected accordingly.

Line 48: change to:

"WOD represents an international collaborative effort among..."

The text has been corrected accordingly.

Lines 56-57: change from "Without any measurements nearby..." to

"For regions without any nearby measurements..."

The text has been corrected accordingly.

Line 58: change to

"… the OI method will underestimate the declining trend…"

 The text has been corrected accordingly.

Line 76:  change "to valid the model" to

"For the evaluation of the model…"

 The text has been corrected accordingly.

Line 175: change "reconstructed model output" to

"…reconstructions fr subsampled model output"

 The text has been corrected accordingly.

Line 177: change "…outputs for the historical simulations to…" to

"…outputs for their historical simulations to…"

 The text has been corrected accordingly.

Line 179: change "The bilinear" to "A bilinear"

 The text has been corrected accordingly.

Line 181: Change "…as the observations" to "..as with the observations"

 The text has been corrected accordingly.

Line 198: Change "…similar to previous studies…" to "..similar to those in previous studies…"

The text has been corrected accordingly.

Line 203: change "…in the south of Greenland" to "…to the south of Greenland"

The text has been corrected accordingly.

Line 330: change "There are two regions, the Subpolar and Subtropical…" to "There are two regions, namely the Subpolar and Subropical…"

The text has been corrected accordingly.

Line 332: change "bracket the observation where some models" to "bracket the observations whereas some models…"

The text has been corrected accordingly.

Line 341: change "indicating that the OI method" to "indicating where the OI method"

The text has been corrected accordingly.

Line 386: replace "gap-filling method used" with "gap-filling method used with observations"

The text has been corrected accordingly.

Line 394: replace "least square sense assuming the Gaussian" with "least squares sense assuming a Gaussian…"

The text has been corrected accordingly.

Line 396: replace "wide-spread" with "widespread"

The text has been corrected accordingly.

Lines 425-426:  replace "at the spacial scales of 10-100km and timescales of several months" with

"With characteristic spatial scales of 10-100km and characteristic timescales…"

The text has been corrected accordingly.

**Reviewer 2**

Ito et al. provide a good try to use synthetic data to understand an objective interpolation method (i.e. Ito et al. approach). They find that the global O2 change might be under-estimated using Ito's objective interpolation approach, which makes sense because the approach infills climatology (no data, no signal) into data-sparse regions. The use of synthetic data is also a good approach I believe. Generally, the paper can be published. But I have some major concerns, mainly on the interpretation and presentation of the results. I hope these concerns can be addressed before publication.

We appreciate the reviewer's support for our use of synthetic data to evaluate a statistical gap-fill approach. Our mapping approach is admittedly simple with a Gaussian covariance function with a constant e-folding scale. This choice has certain benefits, for example, that the results from a simple method are easy to understand, and that it is also easy to notice and to correct mistakes. It can be replicated by other groups relatively easily. If the ocean deoxygenation has a wide-spread, large-scale signal as well as regional hotspots, we anticipate that a simple method should, at least, capture majority of the large-scale component and some regional features. There are some drawbacks too, as correctly pointed out by the reviewer #2. It tends to smooth out spatial gradients, and it may not represent regional signals very well. It is expected that it will underestimate the signal in data poor regions, which we are trying to quantify in this work. Overall, we are in the same opinion with the reviewer #2 that the

revised manuscript should clearly state the limitations of this study and the potential for improvements in future studies.

**Major:**

1. The first concern is that the results are all specific to the particular OI approach proposed by Ito et al., and can not be generalized to all OI approach. This is very important because different groups (even all using OI) have different settings and considerations/assumptions, such as influencing radii, covariance etc., and the performance would be fundamentally different. Therefore, I would strongly insist on being more specific in the paper title and in the Abstract that the "underestimation" is for Ito et al. OI approach.

   We agree that the results presented in this study is based on a specific implementation of the optimal interpolation method. This single study cannot be generalized to all variants of optimal interpolation approaches. As outlined below, the concern can be addressed in the abstract and as well as some modification to the title as suggested by the reviewer #2.

   The tile is revised from "Underestimation of global O2 loss in optimally interpolated historical ocean observations" (original) to "Underestimation of multi-decadal O2 loss due to an optimal interpolation method" (new). The new title is more specific and clearly states that this paper is about a specific optimal interpolation method and it may not be applicable to all other optimal interpolation.

   The revised abstract reinforces the same point. In line 23, the sentence, "an optimal interpolation method is applied to fill data gaps." (original) is revised to "the data gaps are filled by a simple optimal interpolation method using Gaussian covariance with a constant e-folding length scale." (new). Similarly, in line 25 and 27, "the optimal interpolation" (original) is revised to "the simple optimal interpolation" (new). Also in line 29, "Optimal interpolation of the historical dataset estimated the global oxygen loss of 1.5% over the past 50 years. " (original) is revised to "The application of the simple optimal interpolation method to the historical dataset estimated the global oxygen loss of 1.5% over the past 50 years".

2. The sub-sampling strategy has to be more clearly introduced, e.g. how do you re-sample data if a 1X1 grid box has more than 1 observation? How do you deal with the difference in land-ocean masks (models differ from the real-world, for sure)? How do you construct the climatology: are you doing this based on re-sampled data over which time period?

In the original manuscript, there was only one sentence explaining our sampling strategy in L189-191, which was not enough. It would be good to expand and clarify the points raised by the reviewer #2. Below are our answers to the specific questions.

The sub-sampling strategy assumes that a grid box is sampled if, at least, 1 observation exists within the grid cell at a particular year/month. If so, we retain model data in the sub-sampled dataset. In reality, there could be multiple samples within the same grid and/or there could be significant variability in oxygen within the same grid box and within the same year/month. Multiple casts and/or variability within a single cell are not represented in our sub-sampling strategy.

This is now clearly stated in the revised manuscript line 200-206, "The sub-sampling strategy assumes that a grid box is sampled if, at least, 1 observation exists within the grid cell at a particular year/month. If so, we retain model data in the sub-sampled dataset. In reality, there could be multiple casts within the same grid and the same year/month but multiple samples and/or variability within a single cell are not considered. ".

There are slight differences in the land-ocean masks between models, and we use the model topography as they are provided. This could cause some small discrepancies in the ocean volume, but we do not make correction for this specific effect. In revised manuscript line 206-207, this is clearly stated as "There are slight differences in the land-ocean masks between models, and we use the model topography as they are provided."

The analysis period is 1965-2014, and this was already stated in the original manuscript. The climatology is constructed for this period. Sampling pattern affects the representation of climatology, and this effect is included in our analysis, as already stated in the original manuscript. The models' climatological O2 fields are separately calculated for full and sub-sampled model data. These different climatological O2 fields are used to define anomalies for the full and sub-sampled models. Again, these facts are already described in detail in the original manuscript, section 2.4.

3. The over-simplification of the approach has to be stated clearly in the abstract: e.g. no sub-grid variability is considered (observations do have all scales of variability, but your resampled profiles only contain variability >1 grid and 1 month), and no instrumental errors and potential biases are considered. Thus, this current approach is more "conceptual" than you can definitely quantify the

"underestimation". I believe the current study is over-confident in the quantitative numbers of the overestimation. I strongly suggest the authors take this more seriously.

We appreciate the reviewer #2's concern and expanding the points raised in the first comment. The concern of over-generalization is well taken, and it can be addressed primarily in the title and abstract. For our detailed responses, please check the earlier responses.

The sub-grid scale variability that are not represented by earth system models could exist in the real observations, and our current analysis lacks this variability. If sub-grid scale variability were included, it would lead to a lower signal-to-noise ratio and a larger uncertainty. We are aware of this issue and our viewpoint is explained in the sentences starting L562 where we discussed the lack of mesoscale eddies in the model, and we revised the discussion to include the instrumental error as suggested by the reviewer.

We are in agreement with the Reviewer #2 that our model-based analyses lack some sources of variability, and that our analysis might be too optimistic to capture the signal due to the lack of such variability in the models we used. We indeed take this issue seriously, and have suggested some possible solution for future studies in the discussion section.

4. Comparison between model results, and subsampled results with "Observations of Lto et al" (black line in Fig. 4) should be more careful. The so-called "observation" itself is biased by the conservative error (as investigated in this study) and it is more problematic because it also contains sub-grid variability and instrumental problems. Thus, please to be more careful to compare the models and subsampled results with this observation. It makes more sense to remove the black line in Fig.4, and also combines the right and left columns (e.g. full-model in solid lines and subsampled models in dashed lines at the same panels so they can be directly compared). So the focus is on the sampling issue. I have similar concerns about Fig. 5 and 6. I don't even believe "observation" (as you argued in this study, it is problematic in many places because of the sampling/interpolation issues)

These are great suggestions, and we would be very happy to revise Figure 4-6. As suggested, we removed the black line from Fig 4 and we combined left and right column. This change put stronger emphasis on the sampling issue and the comparison between full model and subsampled and gap-filled model output.

The O2 inventory time series based on observations using this simple optimal interpolation method is moved to Figure 3.

Fig 5 and 6 are revised as suggested to provide more detailed spatial structure, and the observation-based map is removed.

5. 7: because of the area difference with latitudes, doesn't it make more sense to define the coverage by area instead of number of cells?

   We appreciate the suggestion. Indeed, it makes more sense to convert the number of cells to areal coverage. Figure 7 and its captions have been updated.

6. 5/6, I would be more keen to see the real spatial pattern (1X1 grid trend), instead of the box averages. It will be more straightforward to show the dynamic regimes.

   We appreciate the suggestion. Fig 5 and 6 are revised as suggested together with the response to the earlier comment #4.

7. 8/9 and section 3.4, with these analyses, I guess the authors want to have an "empirical" correction to the OI approach. But as I said in my 1-3 major points, the quantifications are useless because of the oversimplification of the approach. I don't see the value of doing this analysis.

   We appreciate the reviewer #2's concern. There are indeed limitations in relying on a single, relatively simple implementation of optimal interpolation approach. However, we believe that there still is a value in estimating the bias caused by this simple implementation of optimal interpolation. This type of comparative analysis was never performed collaboratively by modelers and observationalists using synthetic data as a testbed. For the future studies, we would be excited to invite others to join and collaborate including different and perhaps better gap fill approaches.

   Since the primary reason for the objection is mainly comes from the simplicity of our specific gap-fill method, we suggest that it can be addressed with more careful statements about its limitations. This includes revisions in the title, abstract as well as the main text that, this is for a specific, relatively simple implementation of the optimal interpolation. The revisions to the title and abstract are already addressed in the response to comment #1.

   In addition, we have made revision to the maintext. In Line 156 as well as Line 551, "the optimal interpolation" is revised to "a relatively simple optimal

interpolation". In the revised discussion section, this point is better explained from Line 550 to 558.

Overall, I do see the value of this study, but the interpretation and presentation of the results need to be revised in a substantial way to make it a more rigorous study (also leave room for more improvements in the future).

We appreciate the encouragement and support here. There is indeed much room left for more improvements in the future. Potential areas of future improvements are included in the manuscript including Line 550-564, 589-602 in the discussion section.

Minor

1. Abstract "more than 80%", I don't think it is trustable, because the values are apparently model-dependent, and also depend on subgrid-variability and instrumental errors.

   This estimate is subject to the specific implementation of optimal interpolation for this study. As discussed earlier, it will be spelled out more clearly in the revision about its limitation and weakness. In Line 27, "80% of the oxygen trend." (original) is revised to "80% of the oxygen trend in the non-eddying CMIP models". The number could potentially become smaller if we correct the signal-to-noise ratio to a lower value due to unaccounted sub-grid scale variability. It would be important to revisit with more sophisticated estimation approaches in the future study, including the addition of subgrid-variability as discussed in the 4$^{th}$ paragraph in section 4 of the manuscript.

2. Line 105: it is a strange choice, because if you remove Argo because of the precision, then CTD and OSD have different accuracy as well. To me, including Argo is valuable because you just want to test the sampling issue. You do nothing with the precision/accuracy in this paper with a synthetic data approach. I understand the authors might not do the things all over again, so a clear statement on the caveats is a necessary.

   BGC Argo is an important data source especially after mid-2010s. Since the period of trend analysis is from 1967 to 2012, it can impact on the last period. Our decision was to stay on the cautionary side and not to include the ARGO data at this time, but it will be an area for the future study and improvement. A statement is added in the discussion section Line 127. "Float O2 data have been excluded in this study but it will be an important data source especially after 2010s for the future studies."

3. Line 185: just to confirm, is IPSL-CM6A-LR a Earth System Model?

   IPSL-CM6A-LR is the latest version of the IPSL earth system model. In addition to the physical atmosphere-land-ocean-sea ice model based on the LMDz, ORCHIDEE, NEMO (including the LIM and PISCES subcomponents) models. This model includes a representation of the global biogeochemical cycling including carbon, oxygen and nutrients. Further description of the IPSL-CM6A-LR climate model is available through the reference paper listed in Table 1 of the original manuscript.